# Targeting NPC1 in Renal Cell Carcinoma

**DOI:** 10.3390/cancers16030517

**Published:** 2024-01-25

**Authors:** Rushaniya Fazliyeva, Peter Makhov, Robert G. Uzzo, Vladimir M. Kolenko

**Affiliations:** 1Nuclear Dynamics and Cancer Program, Fox Chase Cancer Center, Philadelphia, PA 19111, USA; rushaniya.fazliyeva@fccc.edu; 2Cancer Signaling and Microenvironment Program, Fox Chase Cancer Center, Philadelphia, PA 19111, USA; petr.makhov@fccc.edu; 3Department of Urology, Fox Chase Cancer Center, Philadelphia, PA 19111, USA; robert.uzzo@fccc.edu

**Keywords:** cancer, cholesterol, LDL, HDL, VLDL, NPC1, ccRCC, TKI

## Abstract

**Simple Summary:**

The development of multi-targeted tyrosine kinase inhibitors (TKIs) and immunotherapeutic agents notably changed the treatment paradigm of advanced kidney cancer. However, despite the therapeutic progress, complete and durable responses have been noted in only a few cases. Our studies demonstrate that all major lipoproteins have a comparable ability to supply cholesterol to tumor cells and compromise the antitumor activity of TKIs. Endolysosomal cholesterol transport regulated by NPC1 protein is an attractive therapeutic target based on the fact that this is a point where LDL-, HDL-, and VLDL-derived cholesterol trafficking routes converge and therefore may be simultaneously targeted. Our studies elucidated the role of NPC1 as a potential therapeutic target in clear cell renal cell carcinoma (ccRCC).

**Abstract:**

Rapidly proliferating cancer cells have a greater requirement for cholesterol than normal cells. Tumor cells are largely dependent on exogenous lipids given that their growth requirements are not fully met by endogenous pathways. Our current study shows that ccRCC cells have redundant mechanisms of cholesterol acquisition. We demonstrate that all major lipoproteins (i.e., LDL, HDL, and VLDL) have a comparable ability to support the growth of ccRCC cells and are equally effective in counteracting the antitumor activities of TKIs. The intracellular trafficking of exogenous lipoprotein-derived cholesterol appears to be distinct from the movement of endogenously synthesized cholesterol. De novo synthetized cholesterol is transported from the endoplasmic reticulum directly to the plasma membrane and to the acyl-CoA: cholesterol acyltransferase, whereas lipoprotein-derived cholesterol is distributed through the NPC1-dependent endosomal trafficking system. Expression of NPC1 is increased in ccRCC at mRNA and protein levels, and high expression of NPC1 is associated with poor prognosis. Our current findings show that ccRCC cells are particularly sensitive to the inhibition of endolysosomal cholesterol export and underline the therapeutic potential of targeting NPC1 in ccRCC.

## 1. Introduction

The incidence of kidney cancer has risen steadily over several decades and continues to increase. Renal cell carcinoma (RCC) is the most common form of kidney cancer, whereas clear cell RCC (ccRCC) is the most frequent (75–80%) and the best-studied subtype of RCC. Papillary RCC and chromophobe RCC represent the most common remaining histologic subtypes with an incidence of 7% to 14% and 6% to 11%, respectively [1]. Traditional chemotherapy and radiation therapy are largely ineffective in the treatment of all RCC subtypes [2]. Recently, significant progress in the treatment of advanced ccRCC was achieved with the introduction of targeted agents and checkpoint inhibitors. However, despite the therapeutic progress, complete and durable responses have been noted in only a few cases [3,4,5]. We and others have demonstrated that, in addition to the inhibition of angiogenesis, TKIs also manifest a direct cytotoxic effect on tumor cells [6,7,8,9]. Importantly, TKIs may selectively accumulate in tumor tissue at high concentrations [6,8,10]. Indeed, intra-tumor TKI levels are much higher than peak serum levels (≥10 μM vs. ≤1 μM, respectively) [6,8,10].

ccRCC is a highly lipogenic tumor. The term “clear cell” itself originates from the clear (empty) microscopic appearance of the cytoplasm after the lipids are removed in the process of fixation. ccRCC tissue contains five to eight-fold more total cholesterol than normal kidney tissue [11]. Due to the absolute requirement of cholesterol for the synthesis of cell membranes, rapidly proliferating cancer cells have a greater requirement for cholesterol than normal cells [12,13]. Tumor cells are dependent on exogenous lipids given that their growth requirements are not fully met by endogenous pathways [14]. Also, increased uptake of exogenous cholesterol is preferential for cancer cells compared with time- and energy-consuming de novo cholesterol synthesis. This concept is supported by several studies showing low activity of HMG-CoA reductase (the rate-limiting enzyme in cholesterol synthesis) and reduced cholesterol synthesis in ccRCC cells [11,15].

There are several levels of control in the regulation of cholesterol homeostasis, i.e., cholesterol synthesis, uptake, intracellular trafficking, and efflux. Cholesterol synthesis can be effectively blocked by statins, competitive inhibitors of HMG-CoA reductase. However, cancer cells can bypass the effects of statins by unrestrained cholesterol importation via the LDL receptor (LDLR) pathway [16,17]. These findings provide an explanation for why many tumor cells are resistant to statin treatment. Cholesterol uptake is mediated by a protein transport mechanism. The LDLR supports the efficient uptake of LDL and VLDL [18]. Cholesterol uptake may be also regulated by the VLDL receptor (VLDLR), which shows considerable similarity to the LDLR [15]. The scavenger receptor class B type I (SR-BI) mediates HDL cholesterol uptake [19]. SR-BI can also bind LDL and VLDL, however, less efficiently than LDLR [20]. Up-regulation of SR-BI promotes tumor progression in ccRCC [21]. Lipoprotein-derived cholesterol is delivered to early endosomes. Importantly, the trafficking of exogenous lipoprotein-derived cholesterol appears to be distinct from the movement of endogenously synthesized cholesterol. Newly synthesized cholesterol is transported from the endoplasmic reticulum (ER) directly to the plasma membrane and to acyl-CoA:cholesterol acyltransferase [22], whereas exogenous cholesterol is distributed through the Niemann–Pick type C1 protein (NPC1)-regulated endosomal trafficking system [23,24,25,26,27]. Inhibition of NPC1 causes accumulation of cholesterol in the endolysosomes, a phenotype similar to that observed in Niemann–Pick disease [26,28]. NPC1 plays a critical role in maintaining adequate cholesterol supply in cells that cannot produce endogenous cholesterol [29]. Importantly, the trafficking of de novo synthetized cholesterol in normal cells is not affected by pharmacological or genetic inhibition of NPC1, in contrast to the trafficking of exogenously derived cholesterol [22]. Cellular cholesterol efflux is carried out mainly by ABCA1 and ABCG1 (ATP-binding cassette transporters) [30]. Recent findings enlightened the role of Liver X receptors (LXRs) in cholesterol homeostasis. Activation of LXRs reduces cholesterol uptake by decreasing expression of LDLR and VLDLR [17,31,32,33]. In addition, activation of LXRs stimulates ABCA1-dependent cholesterol efflux [34,35].

Our current studies demonstrate that ccRCC cells depend on the uptake of exogenous cholesterol for their growth and survival and have redundant mechanisms of cholesterol acquisition. We reveal herein that HDL, LDL, and VLDL are equally effective in supplying cholesterol to ccRCC cells and in compromising the antitumor activity of TKIs. Based on our findings, it was anticipated that only concomitant targeting of all sources of cholesterol acquisition or common routes of intracellular cholesterol trafficking would deprive tumor cells of cholesterol supply. We addressed this issue by targeting NPC1-dependent endolysosomal cholesterol transport based on the fact that this is a point where trafficking routes of different lipoproteins converge. Our data show that pharmacological or genetic inhibition of NPC1 reduces viability and sensitizes ccRCC cells to TKIs.

## 2. Materials and Methods

### 2.1. The Cells and Culture Conditions

The 786-O (human ccRCC cell line), RWPE-1, and PZ-HPV-7 (normal prostate epithelial cell lines) were obtained from ATCC (Manassas, VA, USA). PNX0010 cell line was established from a lung metastatic lesion of a ccRCC patient undergoing nephron-sparing surgery at our institution and represents an aggressive TKI-resistant VHL-negative variant of ccRCC [36,37]. SK-RC-45 (human ccRCC cell line), NK680, NK686, NKE (normal kidney epithelial cell lines), and HUVEC (endothelial cell line) were obtained from the Cell Culture Facility (Fox Chase Cancer Center, Philadelphia, PA, USA). Initial stocks were cryopreserved, and at every 6-month interval, a fresh aliquot of frozen cells was used for the experiments. Cells were cultured in RPMI 1640 (Bio-Whittaker, Walkersville, MD, USA) supplemented with 10% FCS (Hyclone, Logan, UT, USA), gentamicin (50 mg/L), sodium pyruvate (1 mM), and non-essential amino acids (0.1 mM) under conditions indicated in the figure legends.

### 2.2. Antibodies and Reagents

LDL, HDL, and VLDL were obtained from Lee Biosolutions (Maryland Heights, MO, USA). Cabozantinib, sunitinib, axitinib, pazopanib, U18666A, and posaconazole were obtained from Cayman Chemical Company (Ann Arbor, MI, USA). Antibody against AR was obtained from Cell Signaling Technology (Danvers, MA, USA).

### 2.3. Western Blot Analysis

Western blot analysis was performed as described previously [38].

### 2.4. Cell Viability and Drug Interaction Analysis

Cell viability was analyzed by CellTiter Blue cell viability assay (Promega, Madison, WI, USA) as described previously [37]. Effective doses (EDs) were calculated using XLfit 2.0, a Microsoft Excel add-in. The synergistic interaction between pharmacological agents was evaluated by the combination index (CI) using CalcuSyn 2.0 software [39]. CI 0.85–0.9: slight synergism; CI 0.7–0.85: moderate synergism; CI 0.3–0.7: synergism; CI 0.1–0.3: strong synergism; CI < 0.1: very strong synergism.

### 2.5. siRNA Transfection

786-O and PNX0010 cells were transfected using SMARTpool siRNA targeting NPC1 (Horizon Discovery, Cambridge, UK, Cat# L-003486-00-0005) essentially as described in our previous report [40].

## 3. Results

### 3.1. LDL, HDL, and VLDL Are Equally Effective in Supporting Viability of ccRCC Cells

Studies by Parinaud et al. demonstrate that all major lipoproteins, (i.e., LDL, HDL, and VLDL) have a similar ability to supply exogenous cholesterol [41]. Based on these findings, we examined the ability of different lipoproteins to support the viability of ccRCC cells. SK-RC-45 [8,42] and PNX0010 [43,44] ccRCC cell lines were cultured in RPMI-1640 medium supplemented with lipid-depleted fetal bovine serum (FBS) in the presence or absence of LDL, HDL, or VLDL. As demonstrated in Figure 1, LDL, HDL, and VLDL were equally effective in maintaining the viability of SK-RC-45 and PNX0010 ccRCC cells cultured under lipid-depleted conditions. The results of this experiment also show that ccRCC cells are largely dependent on exogenous cholesterol supply. In contrast, normal kidney epithelial NK680 cells were less dependent on exogenous cholesterol as indicated by their capacity to maintain viability under lipid-depleted conditions (Figure 1).

### 3.2. LDL, HDL, and VLDL Compromise the Antitumor Activity of TKIs

Our previous findings demonstrate that LDL cholesterol compromises the efficacy of TKIs against ccRCC and endothelial cells [37]. Therefore, we examined whether treatment with HDL and VLDL also affects the antitumor activity of TKIs. Our previous work has shown that TKIs may differ substantially with respect to the mechanism of their antitumor activity [42]. To demonstrate that our observations are not limited to a specific pharmacological agent, we treated SK-RC-45 and PNX0010 ccRCC cells with several clinically relevant TKIs, such as cabozantinib, sunitinib, axitinib, and pazopanib. As shown in Figure 2, LDL, HDL, and VLDL effectively rescued the viability of SK-RC-45 and PNX0010 cells treated with all tested TKIs.

### 3.3. Inhibition of Endosomal Cholesterol Trafficking Sensitizes ccRCC Cells to TKIs

Expression of NPC1 is increased in ccRCC at mRNA and protein levels (Figure 3A,B), and high expression of NPC1 is associated with poor prognosis based on TCGA data analysis (Figure 3C). Given that trafficking of de novo synthetized cholesterol in normal cells is not affected by pharmacological or genetic inhibition of NPC1 [22], we anticipated that ccRCC cells would be particularly sensitive to the inhibition of NPC-1-regulated endosomal cholesterol trafficking. Indeed, ccRCC cells manifested much higher sensitivity to the NPC1 inhibitor U18666A (90) compared with normal cells of various origins (Table 1). Of note, the effect of U1866A and posaconazole, an approved antifungal agent, which directly binds NPC1 and blocks endosomal cholesterol trafficking [45], on tumor cell viability was significantly increased under hypoxia, a central event in renal tumorigenesis (Table 2).

Unobstructed endolysosomal cholesterol trafficking is required to sustain the activity of several signaling pathways that confer drug resistance (i.e., Akt/mTOR, NF-κB, and Erk1/2) [37,46,47,48,49]. Given that, we tested whether NPC1 inhibition enhances the antitumor effect of TKIs. Our experiments reveal that administration of sunitinib, cabozantinib, or pazopanib in combination with U18666A showed a clear synergistic inhibitory effect on the viability of ccRCC cells at all effective dose levels (Table 3).

The results of these experiments were further validated using siRNA-mediated knockdown of NPC1. Genetic depletion of NPC1 reduced viability and sensitized ccRCC cells to TKI treatment (Figure 4 and Table 4).

## 4. Discussion

The lack of sensitivity of ccRCC to chemotherapy and radiation therapy prompted early research efforts into the development of new treatment options. The introduction of TKIs and immune checkpoint inhibitors (ICIs) notably changed the treatment paradigm of ccRCC. Recent clinical studies suggest that combining or sequencing TKIs with ICIs may provide effective treatments that reduce or delay disease progression [50,51,52,53,54,55,56,57]. However, despite the introduction of novel therapeutic approaches in the past decade, advanced ccRCC continues to be a treatment-resistant malignancy [3,4,5].

Multiple studies demonstrate that disruption of cholesterol homeostasis suppresses tumor growth and suggest that targeting cholesterol metabolism may be a promising strategy for antitumor therapy. Indeed, due to the absolute requirement of cholesterol for the synthesis of cell membranes, rapidly proliferating cancer cells have a greater requirement for cholesterol than normal cells (1,2). Our studies demonstrate that ccRCC cells can utilize lipoprotein-derived cholesterol irrespective of the particular class of cholesterol. donor and that all major classes of lipoproteins are capable of compromising the antitumor activity of TKIs. Thus, only simultaneous targeting of different lipoproteins is postulated to have therapeutic benefits. There are several potential approaches for targeting cholesterol homeostasis in cancer cells such as inhibition of cholesterol biosynthesis, blockade of cholesterol uptake, and modulation of intracellular cholesterol trafficking. Increased serum levels of cholesterol in humans can be effectively lowered by PCSK9 inhibitors and statins [58,59,60,61]. PCSK9 is a proprotein convertase that is involved in the degradation of LDLR in the liver [62]. PCSK9 inhibition increases LDLR expression by hepatocytes, which causes increased uptake of circulating LDL, thereby reducing plasma LDL-cholesterol. Statins are competitive inhibitors of HMG-CoA reductase, the rate-limiting enzyme of cholesterol biosynthesis [63,64]. However, these drugs up-regulate HDL levels [60,65,66,67,68], which can serve as a source of cholesterol for tumor cells. Also, statins demonstrate effects on a number of signaling pathways including protein isoprenylation [69]. Anticancer effects of statins have been largely attributed to the inhibition of this post-translational mechanism [69,70]. Moreover, cancer cells can bypass the effects of statins by unrestrained cholesterol importation [16,17]. Given that cholesterol uptake receptors have promiscuous ligand-binding properties, simultaneous targeting of different receptors may be necessary to reduce cellular cholesterol import, which may not be a clinically feasible approach.

As discussed above, tumor cells are largely dependent on exogenous lipids. Endolysosomal cholesterol transport is an attractive therapeutic target based on the fact that this is a point where LDL-, HDL, and VLDL-derived cholesterol trafficking routes converge and therefore may be simultaneously targeted. Furthermore, the trafficking of exogenous lipoprotein-derived cholesterol, in contrast to endogenously produced cholesterol, occurs through an NPC1-mediated endosomal trafficking system [23,24,25,26]. Therefore, the trafficking of de novo synthetized cholesterol in normal cells is not affected by NPC1 inhibition [22]. Indeed, our studies demonstrate that ccRCC cells are particularly sensitive to the inhibition of endosomal cholesterol trafficking by NPC1-specific inhibitor U18666A compared with normal cells of various origins. Importantly, the effect of U1866A and posaconazole on tumor cell viability was significantly increased under hypoxia, a central event in renal tumorigenesis. This observation could be potentially explained by the fact that hypoxia results in the depletion of cholesterol from the plasma membrane [71], increasing the demand for cholesterol transport from an endosomal compartment to the plasma membrane. These findings may widen the therapeutic window of NPC1 inhibitors for more selective targeting of malignant cells.

## 5. Conclusions

Our studies demonstrate that ccRCC cells are highly dependent on the uptake of exogenous cholesterol for their growth and survival and that all major lipoproteins have a comparable ability to supply cholesterol to tumor cells and compromise the antitumor activity of TKIs. Thus, concomitant targeting of all sources of cholesterol acquisition or common routes of cholesterol trafficking may be required to deprive tumor cells of cholesterol supply. Our findings indicate that ccRCC cells are particularly sensitive to the inhibition of NPC-1-dependent endosomal cholesterol trafficking. Importantly, NPC1 expression is increased in ccRCC, and high expression of NPC1 is associated with poor prognosis. Taken together, our work suggests that NPC1 may serve as a potential therapeutic target in ccRCC.

## Figures and Tables

**Figure 1 cancers-16-00517-f001:**
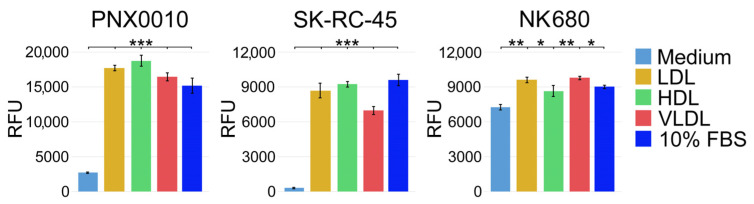
LDL, HDL, and VLDL are equally effective in supporting viability of ccRCC cells. PNX0010 and SK-45 ccRCC cells and NK680 normal kidney epithelial cells were cultured in RPMI 1640 medium supplemented with lipid-depleted fetal bovine serum in the presence or absence of LDL, HDL, or VLDL (all at 100 μg/mL) for 96 h. Cell viability was analyzed by CellTiter Blue assay (Promega). Results are expressed as the mean (n = 3) ± s.e.m. * *p* < 0.01; ** *p* < 0.001; *** *p* < 0.0001. RFU-relative fluorescence units.

**Figure 2 cancers-16-00517-f002:**
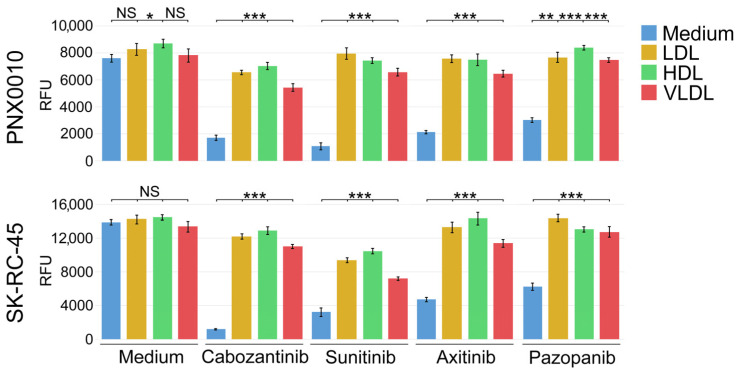
LDL, HDL, and VLDL compromise the antitumor activity of TKIs. SK-45 and PNX0010 ccRCC cells were cultured with TKIs (all at 5 μM) with or without LDL, HDL, or VLDL (all at 100 μg/mL) in RPMI 1640 medium supplemented with regular 10% FBS for 96 h. Cell viability was analyzed by CellTiter Blue assay. Results are expressed as the mean (n = 3) ± s.e.m. * *p* < 0.01; ** *p* < 0.001; *** *p* < 0.0001; NS—Not significant.

**Figure 3 cancers-16-00517-f003:**
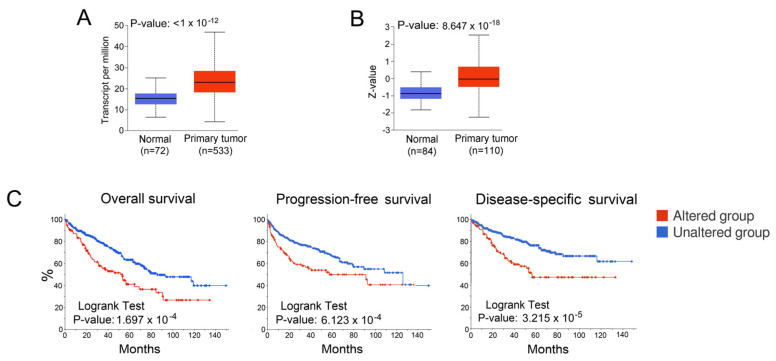
Relationship between NPC1 expression and clinical outcomes in ccRCC. (**A**) Expression of NPC1 is increased in ccRCC at (**A**) mRNA (TCGA) and (**B**) protein (CPTAC) levels according to the University of Alabama at Birmingham cancer data analysis portal (UALCAN). (**C**) NPC1 is an unfavorable prognostic marker in ccRCC based on TCGA analysis via cBioPortal.

**Figure 4 cancers-16-00517-f004:**
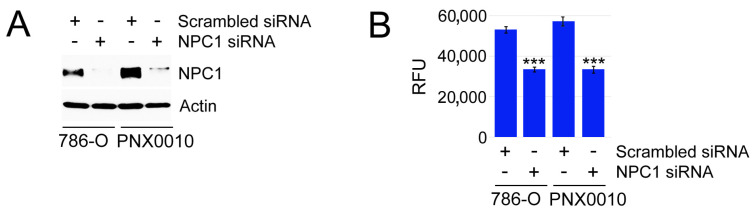
NPC1 depletion reduces viability of ccRCC cells. (**A**) Western blot analysis of NPC1 expression in 786-O and PNX00100 cells treated with control siRNA and NPC1 SmartPool siRNA. The uncropped blots are shown in the Appendix A. (**B**) Aliquots of the cells used in panel A were cultured in RPMI1640 medium supplemented with 10% FBS for 72 h. Cell viability was analyzed as described in legend to Figure 2. *** *p* < 0.0001.

**Table 1 cancers-16-00517-t001:** U18666A preferentially inhibits viability of ccRCC cells. ccRCC (786-O, SK-RC-45, and PNX0010), normal kidney epithelial (NKE and NK686), normal prostate epithelial (RWPE-1 and PZ-HPV-7), and human umbilical vein endothelial (HUVEC) cells were treated with escalating concentrations of U18666A for 72 h. Cell viability and effective doses (ED, μM) were evaluated as described in Materials and Methods.

Cell Line	ED25	IC50	IC75	IC90
786-O	2.1	2.8	3.7	4.9
SK-RC-45	0.09	0.4	1.8	8.2
PNX0010	5.2	10.4	15.9	31.3
NK686	30.7	54.8	97.9	174.7
NKE	26.5	52.9	105.9	211
RWPE-1	21.8	30.3	42.1	58.5
PZ-HPV-7	13.7	28.0	57	116.2
HUVEC	19.7	34.1	59.4	103.6

**Table 2 cancers-16-00517-t002:** The effectiveness of NPC1 inhibitors is enhanced under hypoxic conditions. 786-O and PNX0010 ccRCC cells were cultured under normoxic (N) (20% O_2_) or hypoxic (H) (2% O_2_) conditions in RPMI1640 medium supplemented with 10% FBS and treated with escalating concentrations of either U18666A or posaconazole for 72 h. Cell viability and effective doses (ED, μM) were evaluated as described in Materials and Methods.

Drug	Cell Line	ED25	ED50	ED75	ED90
U18666A	786-O (N)	1.5	2.3	3.5	5.4
	786-O (H)	0.7	1.2	2.1	3.6
	PNX0010 (N)	2.2	3.3	4.9	7.2
	PNX0010 (H)	0.4	0.9	1.9	3.8
Posaconazole	786-O (N)	4.7	10.1	21.6	46.2
	786-O (H)	1.3	3.0	6.7	15.3
	PNX0010 (N)	4.6	8.0	14.2	24.9
	PNX0010 (H)	0.9	1.9	4.2	9.2

**Table 3 cancers-16-00517-t003:** The synergistic effect of combined treatment with U18666A and TKIs on the viability of ccRCC cells. The cells were treated with escalating concentrations of U18666A and TKIs for 72 h. Analysis of the synergistic interaction between U18666A and TKIs was performed as described in Materials and Methods. ED: effective doses. Combination index (CI) 0.85–0.9: slight synergism; CI 0.7–0.85: moderate synergism; CI 0.3–0.7: synergism; CI 0.1–0.3: strong synergism; CI < 0.1: very strong synergism.

Drug	Cell Line	ED25	ED50	ED75	ED90
Sunitinib	786-O	0.56	0.52	0.47	0.41
	SK-RC-45	0.45	0.41	0.37	0.32
	PNX0010	0.50	0.46	0.42	0.39
Cabozantinib	786-O	0.44	0.40	0.35	0.30
	SK-RC-45	0.39	0.33	0.27	0.22
	PNX0010	0.52	0.45	0.41	0.38
Pazopanib	786-O	0.37	0.31	0.25	0.20
	SK-RC-45	0.42	0.36	0.29	0.23
	PNX0010	0.48	0.40	0.33	0.26

**Table 4 cancers-16-00517-t004:** Depletion of NPC1 sensitizes ccRCC cells to TKIs. Control and NPC1-depleted cells described in Figure 4 were treated with escalating concentrations of axitinib and cabozantinib for 72 h. Viability and EDs (μM) were calculated as described in the legend in Table 1.

Drug	Cell Line	ED25	ED50	ED75	ED90
Axitinib	786-O	5.4	7.4	10.1	13.9
	786-O-NPC1^KD^	1.3	2.5	5.0	9.9
	PNX0010	4.1	6.9	11.5	19.3
	PNX0010-NPC1^KD^	1.5	2.6	4.7	8.5
Cabozantinib	786-O	5.6	8.5	12.8	19.3
	786-O-NPC1^KD^	2.5	4.0	6.4	10.4
	PNX0010	4.6	6.8	10.0	14.9
	PNX0010-NPC1^KD^	2.7	3.9	5.7	8.2

## Data Availability

The data presented in this study are available from the corresponding author (V.M.K.) upon request.

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
