# Peer review of "Targeting NPC1 in Renal Cell Carcinoma"

_cancers, 2024, doi:10.3390/cancers16030517_

Round 1

Reviewer 1 Report

Comments and Suggestions for Authors

18 December 2023

Ms. Ref. No.: cancers-2794469

Journal: Cancers

Title: Targeting NPC1 in renal cell carcinoma

Comments:

Thank you for your efforts in writing this article on a very pertinent topic. Moreover, I found the article to be informative and with the potential for further research on this topic in future.

I have some observations where mentioned in the following paragraphs that will be useful for its improvement:

1-      One of the aims of this study is assessing association of the cholesterol with renal cancer, among a lot of factors of lipid profile, why selecting the cholesterol?

2-      It seems better that the complete of ccRCC will be mentioned in lines 19 (Simple Summary) and 22 (Abstract).

3-      Why two normal cell lines (RWPE-1 and PZ-HPV-7) were used in this research while only one human ccRCC cell line was checked?

4-      According to Figures such as Fig 1, which of these lipoproteins LDL, HDL, and VLDL has the main role in this renal cancer? And why?

5-      What was the time rang of cell viability test? In the captions of figures and tables mentioned 72 h and mentioned that Cell viability and effective doses were evaluated as described in Materials and Methods. Additionally please reintroduce the condition of these tests.

6-      Moreover, The Following reference can be included in the introduction part for more readability:

·         https://doi.org/10.3390/cancers15235637   

·         https://doi.org/10.1007/s00210-023-02551-0  

·         https://doi.org/10.1007/s11033-021-06928-3

·         https://doi.org/10.1007/s11033-021-06928-3

Author Response

  1. “One of the aims of this study is assessing association of the cholesterol with renal cancer, among a lot of factors of lipid profile, why selecting the cholesterol?”

Due to the absolute requirement of cholesterol for the synthesis of cell membranes, rapidly proliferating cancer cells have greater requirement for cholesterol than normal cells. ccRCC cells are auxotrophic for exogenous cholesterol (Riscal et al. Cholesterol Auxotrophy as a Targetable Vulnerability in Clear Cell Renal Cell Carcinoma. Cancer Discov. 2021 Dec 1;11(12):3106-3125). Our current and previous studies (Naito et al. LDL cholesterol counteracts the antitumour effect of tyrosine kinase inhibitors against renal cell carcinoma. Br J Cancer. 2017 Apr 25;116(9):1203-1207) support these findings and demonstrate that depriving ccRCC cells of cholesterol inhibits their proliferation and reinstates sensitivity to tyrosine kinase inhibitors (TKIs). Therefore, cholesterol homeostasis is an attractive therapeutic target in renal cancer.

  1. “It seems better that the complete of ccRCC will be mentioned in lines 19 (Simple Summary) and 22 (Abstract).”

This comment is appreciated. We spelled out ccRCC abbreviation in Simple Summary and Abstract as suggested by the expert reviewer.

  1. “Why two normal cell lines (RWPE-1 and PZ-HPV-7) were used in this research while only one human ccRCC cell line was checked?”

We used three ccRCC cell lines in our experiments, i.e. 786-O, SK-RC-45 and PNX0010.  In the experiments presented in Table 1, we also used two human normal kidney epithelial cell lines (NK686 and NKE), two human normal prostate epithelial cell lines (RWPE-1 and PZ-HPV-7) and human umbilical vein endothelial cell line (HUVEC) to demonstrate that ccRCC cells are more sensitive to the inhibition of endosomal cholesterol transport than normal cells of various origins.

  1. “According to Figures such as Fig 1, which of these lipoproteins LDL, HDL, and VLDL has the main role in this renal cancer? And why?”

The results presented in our manuscript, including data shown in Figure 1, indicate that ccRCC cells have redundant mechanisms of cholesterol acquisition and can utilize lipoprotein-derived cholesterol irrespective of the particular class of cholesterol donor to ensure its adequate supply. Thus, all major lipoproteins (i.e., LDL, HDL, and VLDL) have a similar ability to support growth of ccRCC cells and are equally effective in counteracting the antitumor activities of TKIs.

  1. “What was the time range of cell viability test? In the captions of figures and tables mentioned 72 h and mentioned that Cell viability and effective doses were evaluated as described in Materials and Methods. Additionally, please reintroduce the condition of these tests.”

Viability of cells cultured in the presence of LDL, HDL, and VLDL (Figures 1 and 2) was examined 96 hours following treatment initiation. Viability of cells treated with pharmacological agents (Tables 1-4) or transfected with NPC1 SmartPool siRNA (Figure 4) was examined 72 hours following treatment initiation. This information is included in figure and table legends.

  1. “Moreover, the Following reference can be included in the introduction part for more readability: https://doi.org/10.3390/cancers15235637 

https://doi.org/10.1007/s00210-023-02551-0 

https://doi.org/10.1007/s11033-021-06928-3”

Sorry, but none of these references are relevant to the topic of our manuscript “Targeting NPC1 in Renal Cell Carcinoma”. Please see below:

https://doi.org/10.3390/cancers15235637 

“Inflammation and Immunity Gene Expression Patterns and Machine Learning Approaches in Association with Response to Immune-Checkpoint Inhibitors-Based Treatments in Clear-Cell Renal Carcinoma.”

https://doi.org/10.1007/s00210-023-02551-0 

“The anti-tumoral role of Hesperidin and Aprepitant on prostate cancer cells through redox modifications.”

https://doi.org/10.1007/s11033-021-06928-3

“Potential in vitro therapeutic effects of targeting SP/NK1R system in cervical cancer.”

Reviewer 2 Report

Comments and Suggestions for Authors

·         A brief summary: Cancer cells, especially clear cell renal cell carcinoma (ccRCC), exhibit heightened demand for cholesterol, primarily relying on external lipids due to insufficient endogenous pathways. This study reveals ccRCC's redundant mechanisms for acquiring cholesterol from major lipoproteins, emphasizing the potential of targeting NPC1 for therapeutic intervention in ccRCC by inhibiting endolysosomal cholesterol export.

·         General concept comments:

Review: The research article is well written in fluent English, but I request the authors to address the following comments to have a better understanding of the study and its potent future implications:

1.     The authors assert NPC1 as a potential therapeutic target in ccRCC. I recommend exploring downstream mechanistic effects of NPC1 depletion, specifically its impact on physiological processes like the DNA damage response.

2.     Please investigate whether CRISPR Cas9-mediated NPC1 knockout yields comparable sensitization levels in ccRCC cells and normal cells as observed post its siRNA mediated knockdown, providing insights into potential therapeutic selectivity.

3.     Consider assessing the druggability of NPC1's protein structure through docking simulations to inform the feasibility of targeting it pharmacologically.

4.     Explore combinatorial therapies, assessing synergistic effects of novel NPC1 inhibitors with conventional TK inhibitors against kidney cancer for enhanced therapeutic outcomes.

Author Response

  1. “The authors assert NPC1 as a potential therapeutic target in ccRCC. I recommend exploring downstream mechanistic effects of NPC1 depletion, specifically its impact on physiological processes like the DNA damage response.”

NPC1 loss-of-function in humans leads to the Niemann-Pick type C (NPC) disease, a childhood-onset neurodegenerative disorder characterized by intracellular lipid accumulation leading to cell death and organ damage. Cellular and pathophysiological consequences of NPC disease are well-described in the literature (reviewed in Antioxidants (Basel). 2023 Dec; 12(12): 2021; J Neurochem. 2020 Jun;153(6):674-692; Handb Clin Neurol. 2013:113:1717-21). Also, our current manuscript was submitted in response to the invitation to a Topical Collection "Clear Cell Renal Cell Carcinoma 2022-2023". Therefore, our studies focused on elucidating the role of NPC1 as a potential therapeutic target in ccRCC.

  1. “Please investigate whether CRISPR Cas9-mediated NPC1 knockout yields comparable sensitization levels in ccRCC cells and normal cells as observed post its siRNA mediated knockdown, providing insights into potential therapeutic selectivity.”

CRISPR/Cas9-mediated knockdown typically requires three-to-four-week long antibiotic-based selection of CRISPR/Cas9 edited cell pools (Stoyko et al. CRISPR-Cas9 Genome Editing and Rapid Selection of Cell Pools. Curr Protoc. 2022 Dec; 2(12): e624). Cells auxotrophic for cholesterol experience progressive loss of viability following 3-4 days of culture under compromised cholesterol supply (O’Neill et al. NPC1 Confers Metabolic Flexibility in Triple Negative Breast Cancer. Cancers (Basel). 2022 Jul; 14(14): 3543). ccRCC cells are functional auxotrophs for exogenous cholesterol and undergo cell death in its absence (Riscal et al. Cholesterol Auxotrophy as a Targetable Vulnerability in Clear Cell Renal Cell Carcinoma. Cancer Discov. 2021 Dec 1;11(12):3106-3125). NPC1 plays a critical role in cellular homeostasis of exogenous cholesterol. This cellular phenotype would compromise our ability to perform these studies making them not technically feasible. In our manuscript, we demonstrated the outcomes of NPC1 suppression using genetic (siRNA) and pharmacological (U18666A) approaches.

  1. “Consider assessing the druggability of NPC1's protein structure through docking simulations to inform the feasibility of targeting it pharmacologically.”

3D structure of NPC1 has been reported previously (Li et al. Structure of human Niemann-Pick C1 protein. Proc Natl Acad Sci U S A 2016 Jul 19;113(29):8212-7; Li et al. A structure of Niemann-Pick C1 protein reveals insights into the function of the C-terminal luminal domain in cholesterol transport. Proc. Natl. Acad. Sci. USA 114, 9116–9121 (2017); Li et al. Clues to the mechanism of cholesterol transfer from the structure of NPC1 middle lumenal domain bound to NPC2. Proc. Natl. Acad. Sci. USA 113, 10079–10084 (2016)). Furthermore, recent studies identified structural basis for pharmacological NPC1 inhibition by triazoles (Long et al. Structural basis for itraconazole-mediated NPC1 inhibition. Nat Commun. 2020 Jan 9;11(1):152) and NPC1-specific inhibitor U18666A (Li et al. Structure of human Niemann-Pick C1 protein. Proc Natl Acad Sci U S A 2016 Jul 19;113(29):8212-7). Our current studies demonstrate significant antitumor activity of posaconazole and U18666A against ccRCC cells.

  1. “Explore combinatorial therapies, assessing synergistic effects of novel NPC1 inhibitors with conventional TK inhibitors against kidney cancer for enhanced therapeutic outcomes.”

Our experiments (Table 3) show the prominent synergistic effect of combined treatment of ccRCC cells using NPC1-specific inhibitor U18666A and several clinically relevant TKIs, such as cabozantinib, sunitinib and pazopanib.

Round 2

Reviewer 1 Report

Comments and Suggestions for Authors

no thanks